# Correspondence: Uncertainty in Climate-Vegetation Feedbacks on Fire Regimes Challenges Reliable Long-Term Projections of Burnt Area from Correlative Models

**Lluís Brotons** [1,2,3,*] and **Andrea Duane** [1,2]

1. InForest jru (CTFC-CREAF), Solsona 25280, Spain
2. CREAF, Cerdanyola del Vallès 08193, Spain; andrea.duane@ctfc.cat
3. CSIC, Cerdanyola del Vallès 08193, Spain
*   Correspondence: lluis.brotons@ctfc.cat

**Abstract:** Recent studies have explored the use of simple correlative models to project changes in future burnt areas (BAs) around the globe. However, estimates of future fire danger suffer from the critical shortcoming that feedbacks on climate change effects on vegetation are not explicitly included in purely correlative approaches causing potential major unknown biases on BA projections. In a recent application of this approach led by Marco Turco and co-workers in the journal Nature Communications (doi:10.1038/s41467-018-06358-z), a simple correlative model was used to project an increase in future burnt areas for the Mediterranean region. The authors related BAs to regional estimates of cumulative drought surrogates, and later used this relationship to infer changes derived from future climate data. To account for negative climate-vegetation feedback on fire regimes, they used regional variability in the BA–drought relationship. The main assumption behind the approach used was that fire–drought relationships currently measured under a given productivity gradient (i.e., sensitivity of fire activity to dry periods is stronger in cooler/productive sites) can be consistently used to infer new relationships arising in the future. While representing a step forward in acknowledging the pitfalls of current projections of BAs, this short-cut falls short in allowing to account for the key process behind climate–vegetation-fire feedbacks. We argue that a series of mechanisms, ranging from the dynamic nature of fire–drought relationships to the human influences they experience, do not ensure that these relationships are to be maintained in the future with major, overall still unknown, consequences on future fire danger projections. Resolving this challenge will greatly benefit from the development of mechanistic approaches that explicitly consider the processes by which vegetation changes derived from climate influence fire regimes.

**Keywords:** fire; drought; correlative models; Mediterranean systems; climate change; socio-ecological systems; tipping points

---

Recent studies have explored the use of simple correlative models to project changes in future burnt areas (BAs) around the globe [1,2]. In a recent application of this approach, Turco et al. (2018) [3] published an elegant work in which they apply a simple correlative model to project an increase in future burnt areas (BAs) for the Mediterranean region. In their study, the authors relate BAs to regional estimates of cumulative drought surrogates derived from evapotranspiration indices, and later they use this relationship to infer changes derived from future climate data. However, estimates of future burnt areas suffer from the critical shortcomings that positive and negative feedbacks of climate change on vegetation (i.e., climate may actually reduce vegetation growth and eventually decrease fire

danger) are not included [2]. To overcome this problem, a way around this shortcoming is proposed by using regional variability in the BA–drought relationship (what they call non-stationary models) to account for future changes in fire regimes derived from climate effects on vegetation. Their analyses showed that the sensitivity of fire activity to dry periods is stronger in cooler/productive sites and therefore, they propose to use this finding as a short cut in their BA projections using climate change scenarios. The main assumption behind this approach is that the fire–drought relationships under a given productivity gradient can be used to infer the new relationships arising in the future. While representing a step forward in acknowledging the pitfalls of current BA projections, this short-cut falls short in allowing to account for the key process behind climate–vegetation-fire feedbacks. We argue that there are a series of mechanisms by which current correlations are not likely to be maintained in the future with major, overall still unknown, consequences on BA projections.

First, fire–drought relationships are affected by ecosystem productivity [4,5]. However, climate drivers of productivity are not universal, and often difficult to capture [6]. In their study, Turco et al. [3], found that temperature proved to be the best predictor of regional variability in the fire–drought relationship as estimated by the association between BAs and evapotranspiration indices. We appeal that temperature might not be the most appropriate variable to relate productivity changes in vegetation to fire activity in the Mediterranean [7]. For instance, Pausas and Ribeiro [4] pointed to an increase in net primary productivity across the range of temperatures included in Turco et al. (2018) [3], suggesting that other factors beyond productivity may be behind the relationships found by the later. Current evidence suggests that vegetation productivity in the Mediterranean is usually constrained by precipitation rather than by temperature alone [8]. In line with this, the authors actually conducted a sensitivity analysis to account for uncertainty derived from different climate variables potentially related to productivity gradients. Then, by using a wider set of predictors including precipitation related variables, the models lead to an enormous increase in the uncertainty of BA predictions (BA increase of up to ~250% in the non-stationary model built using water balance related variables; Figure 5 from [3]). Results suggest that these correlative models are extremely idiosyncratic in nature and that using them for future projections is, at present, extremely uncertain. In addition, the sensitivity of vegetation to fire is not always mediated by productivity. Wind-driven fire regime regions are likely to strongly affect these relationships by reducing the role of vegetation in fire spread patterns [9,10]. By not including this critical climate factor in their correlative analyses, BA drought analyses may be biased, especially in regions in where these two climate characteristics co-occur.

Second, spatial variation in the fire–drought relationship is likely to account for the effects of lower vegetation growth in drier sites, which erodes the capacity of fuel to build up rapidly and thus reduces the dominance of larger fires [7]. It is our view that using the current estimated BA-evapotransporation coefficients as a shortcut falls short in accounting for the interactions we expect to drive future climate–fire-vegetation feedbacks. The direct application of fire–drought relationships from other regions would assume a complete shift in vegetation, which is far from likely in the forthcoming years due to lags in vegetation responses to climate [11]. Under this view, we expect that the approach used by Turco et al. [3], based on nonstationary models, would actually shift BA values closer to those obtained in their stationary models. However, since climate conditions anticipated in the future are likely to be novel in many cases, vegetation responses are highly uncertain and may involve more sudden, case specific changes falling outside the variability estimated in the paper. In these cases, large scale vegetation shifts (i.e., *Pinus* to *Quercus* transitions [12]) triggered by the fire itself, climate, or a combination of both [13] may increase the sensitivity of the system to drier conditions even faster than what is assumed in the paper. Under this contrasting view, we could expect in fact lower BA projections compared to what is expected from nonstationary models. It is worth noticing that recent sensibility analyses comparing correlative and mechanistic models carried out in Western USA actually support the view that under particular conditions, and when accounting for negative climate–vegetation feedbacks, BA projections can in fact decrease in comparison with current figures [2]. Overall, assumptions on specific vegetation responses to climate productivity changes for

which mechanisms are not yet fully understood are likely to induce contradictory effects on future estimate of BAs.

Finally, climate–fire-vegetation feedbacks are often mediated by human factors. This human influence is likely to be especially strong for European Mediterranean forest systems [14]. Although this is explicitly recognized by Turco et al. [3], we think their projections may strongly suffer from biases associated to these human influences [15]. In contrast with the future projections showing increases in BA, these authors have actually reported significant negative recent trends [3] (mainly derived from fire suppression) in BAs independent of the fire–drought relationship. This indicates that contrary to the assumption that BAs will be affected by spatial variation in drought alone, BAs are likely to be heavily influenced by other factors with variability in ignition patterns or fire suppression policies being outstanding examples. While Turco et al. [3] argue that their approach does not account for future changes in fire management policies, we argue that these fire management policies are already shaping the fire–drought relationships in the region and therefore may strongly influence projections derived from these simple relationships [16].

While we share some of the worries raised by the authors [3] in relation to the increase in climate fire-prone conditions under even mild increases in temperature (which goes in line with evidence already provided by a number of studies [17,18]), we argue that purely correlative approaches currently available cannot adequately capture vegetation feedbacks expected from fire regime changes. Their results entail a large uncertainty preventing reliable inference on long-term projections of BAs. Current evidence cannot yet discard that negative feedbacks actually decreases burnt areas in the long term in some regions specially when combined with high fire suppression effectiveness. Resolving this challenge will greatly benefit from the development of mechanistic models or approaches that explicitly consider the processes by which changes on vegetation derived from climate influence fire regimes [19]. Future research focus in modeling approaches will also require a more in depth inclusion and assessment of novel fire conditions (and especially novel wind patterns), and their relationships with vegetation responses and fire suppression effectiveness in determining future fire regimes in the region.

**Funding:** This research was funded by the Spanish Ministry of Economy, Industry and Competitiveness (INMODES project CGL2017-89999-C2-2-R). Additional funding came from the European Commission via the Marie Curie RISE project SUFORUN (H2020-MSCA-RISE-2015) and the *Generalitat de Catalunya* (CERCA Program). Andrea Duane was funded by the *Ministerio de Educación, Cultura y Deporte* (Spain) (FPU13/00108) and (EST16/00984).

**Conflicts of Interest:** The authors declare no conflict of interest. The funders had no role in the design of the study; in the collection, analyses, or interpretation of data; in the writing of the manuscript, or in the decision to publish the results.

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
