# Peer review of "Correspondence: Uncertainty in Climate-Vegetation Feedbacks on Fire Regimes Challenges Reliable Long-Term Projections of Burnt Area from Correlative Models"

_fire, doi:10.3390/fire2010008_

Round 1
Reviewer 1 Report
The comment manuscript by Brotons and Duane points out one main shortcoming of a recent paper by Turco et al. published in Nature Communications.
Turco et al. projected future burnt areas (BA) in Mediterranean Europe at a regional level (NUTS3). Their methodology, quite simple, is based on the statistical relationship (linear regression model) between yearly BA and the Standardized Precipitation Evaporation Index (SPEI), determined from historical observations for each spatial unit. Those relationships are then used to project BA under different climate scenarios and evaluate some of the uncertainty sources of BA projections. One interesting point in Turco et al. (2018), and that is the subject of the comment by Brotons and Duane, is to explicitly consider the impacts of the modifications in the BA-SPEI relationship that might result from productivity alterations induced by climate changes, by using what they called ‘non-stationary models”. To that end, the regression coefficient (β2) of the BA-SPEI relationships is correlated to mean temperature (on a spatial basis) and the β2-Temperature relationships if subsequently used in projections to alter BA sensitivity to SPEI.
In their short comment, Brotons and Duane specifically call into question the rationale behind these non-stationary models as well as the interpretability and added value of BA projected using this methodology. They argue that “the approach used by Turco et al. cannot adequately capture vegetation feedbacks expected from fire regime changes”. This is well written manuscript with sound arguments, which I read with great interest. This manuscript manages to stimulate the scientific debate without being too harsh towards the author of the Turco’s study, which is not always easy in that kind of exercise. Here I must say that I also was one reviewer of the paper by Turco, which surely explain that I feel particularly concerned by this comment. In this regard, the review of the Turco’s paper, which is publicly available, might also be of interest (see at the following link: https://static-content.springer.com/esm/art%3A10.1038%2Fs41467-018-06358-z/MediaObjects/41467_2018_6358_MOESM2_ESM.pdf). Nevertheless, it seems that the Turco’s paper has raised some debates within the European and Mediterranean fire science community. This comment therefore came as no surprise and I am sure that it will be of interest for the readers of the journal and wildfire scientists.
Despite these strengths, my impression is that this manuscript could be substantially improved by thoroughly reorganizing the argumentation. In its current state, it consists of three distinct paragraphs, but I found that main idea argued in each of sections was not clear and several similar ideas can be found in these different paragraphs, which clearly prevents from a quick understanding of the author’s arguments and rationale. Symptomatically, at no point in the manuscript did the authors sum up their main ideas (not even in the abstract). I also had hard time trying to sum-up the main author’s ideas for the purpose of this review and came to conclusion that the authors choice regarding the organization of the manuscript was maybe not optimal. The way I see it, section 1 (L49-69) and 3 (L86-L97) could be merged in one section questioning the rationale and what really lies behind β-Temperature regressions models. A second section could address the rationale for using these relationships as a proxy of productivity alterations for future BA projections and would take up most ideas already developed in the second paragraphs (L69-L85). Of course, this is only a suggestion, but I strongly recommend the authors to rewrite their manuscript around 2 or 3 main ideas to increase its clarity and impact.
Other comments
# Maybe one additional point that was not directly raised by the Brotons and Duane also relates directly to the validity of the drought-fire activity relationship in the fuel vs drought limited framework. Namely, there are reasons to believe (e.g. Batllori et al. 2013) that some regions (at the southern edge of the productivity gradient) will shift from drought to fuel limitation, which would also question the validity of the current BA-SPEI and β-Temperature relationships. This possibility is quickly excluded in Turco et al. (2018) by a sentence in the introduction that reads : « Only if the direct effect of climate change in regulating fuel moisture (e.g., drier and warmer conditions increase fuel flammability leading to larger fires) continues to be dominant with respect to the indirect effect on fuel load and structure (e.g., drier and warmer conditions limit fuel availability), fire risks will increase7,9,29–32 as the climate becomes warmer and drier33,34 ».
# I also recommend the authors to consider the following paper that specifically deals with some of the issues discussed in their manuscript.
Harris, R. M. B., Remenyi, T. A., Williamson, G. J., Bindoff, N. L., & Bowman, D. M. J. S. (2016). Climate–vegetation–fire interactions and feedbacks: trivial detail or major barrier to projecting the future of the Earth system? Wiley Interdisciplinary Reviews: Climate Change, 7(6), 910–931. https://doi.org/10.1002/wcc.428
Technical comments
# “fire risk” is sued several times in the manuscript but this expression is usually employed to also include also the values at stake. You should consider replace it by “fire danger” or “BA projections”.
# L55-56: Please rephrase
# L60: precipitatioN
# L66: PatterNs
# L71: IT is our view…
# L89-91: Please provide the reference
Author Response
Reviewer 1
The comment manuscript by Brotons and Duane points out one main shortcoming of a recent paper by Turco et al. published in Nature Communications.
Turco et al. projected future burnt areas (BA) in Mediterranean Europe at a regional level (NUTS3). Their methodology, quite simple, is based on the statistical relationship (linear regression model) between yearly BA and the Standardized Precipitation Evaporation Index (SPEI), determined from historical observations for each spatial unit. Those relationships are then used to project BA under different climate scenarios and evaluate some of the uncertainty sources of BA projections. One interesting point in Turco et al. (2018), and that is the subject of the comment by Brotons and Duane, is to explicitly consider the impacts of the modifications in the BA-SPEI relationship that might result from productivity alterations induced by climate changes, by using what they called ‘non-stationary models”. To that end, the regression coefficient (β2) of the BA-SPEI relationships is correlated to mean temperature (on a spatial basis) and the β2-Temperature relationships if subsequently used in projections to alter BA sensitivity to SPEI.
In their short comment, Brotons and Duane specifically call into question the rationale behind these non-stationary models as well as the interpretability and added value of BA projected using this methodology. They argue that “the approach used by Turco et al. cannot adequately capture vegetation feedbacks expected from fire regime changes”. This is well written manuscript with sound arguments, which I read with great interest. This manuscript manages to stimulate the scientific debate without being too harsh towards the author of the Turco’s study, which is not always easy in that kind of exercise. Here I must say that I also was one reviewer of the paper by Turco, which surely explain that I feel particularly concerned by this comment. In this regard, the review of the Turco’s paper, which is publicly available, might also be of interest (see at the following link: https://static-content.springer.com/esm/art%3A10.1038%2Fs41467-018-06358-z/MediaObjects/41467_2018_6358_MOESM2_ESM.pdf). Nevertheless, it seems that the Turco’s paper has raised some debates within the European and Mediterranean fire science community. This comment therefore came as no surprise and I am sure that it will be of interest for the readers of the journal and wildfire scientists.
Despite these strengths, my impression is that this manuscript could be substantially improved by thoroughly reorganizing the argumentation. In its current state, it consists of three distinct paragraphs, but I found that main idea argued in each of sections was not clear and several similar ideas can be found in these different paragraphs, which clearly prevents from a quick understanding of the author’s arguments and rationale. Symptomatically, at no point in the manuscript did the authors sum up their main ideas (not even in the abstract). I also had hard time trying to sum-up the main author’s ideas for the purpose of this review and came to conclusion that the authors choice regarding the organization of the manuscript was maybe not optimal. The way I see it, section 1 (L49-69) and 3 (L86-L97) could be merged in one section questioning the rationale and what really lies behind β-Temperature regressions models. A second section could address the rationale for using these relationships as a proxy of productivity alterations for future BA projections and would take up most ideas already developed in the second paragraphs (L69-L85). Of course, this is only a suggestion, but I strongly recommend the authors to rewrite their manuscript around 2 or 3 main ideas to increase its clarity and impact.
Response: Thanks for the comments. We have rewritten the abstract in order to better summarize the main points raised by our letter. However, after some thought we have opted for not reorganizing the letter. We think it is now consistently structured around three main ideas (1- Uncertainty related to use of different basic drivers of fire-drought relationships, 2- Uncertainty derived from the use of spatial surrogates of the fire-drought relationships and 3- Uncertainty derived from unknown human related factors mediating the relationship) and therefore, a big change in this arrangement would modify its main spirit.
Other comments
# Maybe one additional point that was not directly raised by the Brotons and Duane also relates directly to the validity of the drought-fire activity relationship in the fuel vs drought limited framework. Namely, there are reasons to believe (e.g. Batllori et al. 2013) that some regions (at the southern edge of the productivity gradient) will shift from drought to fuel limitation, which would also question the validity of the current BA-SPEI and β-Temperature relationships. This possibility is quickly excluded in Turco et al. (2018) by a sentence in the introduction that reads: « Only if the direct effect of climate change in regulating fuel moisture (e.g., drier and warmer conditions increase fuel flammability leading to larger fires) continues to be dominant with respect to the indirect effect on fuel load and structure (e.g., drier and warmer conditions limit fuel availability), fire risks will increase7,9,29–32 as the climate becomes warmer and drier33,34 ».
Response: We think this is a very valid point. However, our view is that in fact the rationale used in that sentence in the introduction is part of the set of reasons (discussed in our comment) for which future changes in vegetation make actual burnt area projections an uncertain challenge.
# I also recommend the authors to consider the following paper that specifically deals with some of the issues discussed in their manuscript.
Harris, R. M. B., Remenyi, T. A., Williamson, G. J., Bindoff, N. L., & Bowman, D. M. J. S. (2016). Climate–vegetation–fire interactions and feedbacks: trivial detail or major barrier to projecting the future of the Earth system? Wiley Interdisciplinary Reviews: Climate Change, 7(6), 910–931. https://doi.org/10.1002/wcc.428
Response: Reference added.
Technical comments
# “fire risk” is sued several times in the manuscript but this expression is usually employed to also include also the values at stake. You should consider replace it by “fire danger” or “BA projections”.
# L55-56: Please rephrase
# L60: precipitatioN
# L66: PatterNs
# L71: IT is our view…
# L89-91: Please provide the reference
Response: We have corrected the technical comments outlined by the reviewer.
Reviewer 2 Report
I have reviewed this short commentary and think it well-supported, and it addresses a serious issue in the projections of future burned area. This is an important message to be conveyed. My major suggestion would be that, instead of singling out one study, to acknowledge some of the other studies that have used or discussed SPEI in correlative models to project burned area under climate change (albeit without it being labeled ‘nonstationary’, e.g., Krawchuk et al. 2014, Mann et al. 2016)- and to acknowledge other work that makes similar arguments as here (e.g., Syphard et al. 2018, Bowman et al. 2009). This widens the commentary into a broader statement, and it underlines the importance of it. References at end.
Overall, there are some awkward word choices and redundancy, so another edit for language may be merited. It could help tighten up the wording too.
Minor comments:
L15 – and again in introduction, certainly there are most likely to be negative feedbacks, but there could be other effects or feedbacks, depending on direct and indirect climatically or fire-driven changes due to CO2, direct species distribution changes, species response to fire, etc. Maybe just say “changes in vegetation” here, then explain as you already do in introduction.
Also, just a note, but the abstract and beginning of introduction are almost verbatim the same. Maybe flesh out introduction a bit more, depending on word limits?
L14, It took me a while to stop having to look up what SPEI meant when I read it. Might be more intuitive to just call it “fire-drought” or “fire-evapotranspiration” ?
L31, Perhaps start with a background statement (or s) like in the abstract.
L62, What is PRE-PET?
L86, In this paragraph, which I totally agree with, perhaps mention that human influence is more than just management? fire ignition patterns may also come into play and are not accounted for in the correlative models.
Mann M, Batllori E, Moritz M, Waller E, Berck P, Flint A, et al. Incorporating anthropogenic influences into fire probability models: effects of human activity and climate change on fire activity in California. PLoS One. 2016;11(4):e0153589.
Krawchuk MA, Moritz MA. Burning issues: statistical analyses of global fire data to inform assessments of environmental change. Environmetrics [Internet]. 2014;25(6):472–81.
Bowman DMJS, Balch JK, Artaxo P, Bond WJ, Carlson JM, Cochrane MA, et al. Fire in the Earth System. Science (80-) [Internet]. 2009;324(5926):481–4.
Syphard, Alexandra D., et al. "Mapping future fire probability under climate change: Does vegetation matter?." PloS one13.8 (2018): e0201680.
Author Response
Reviewer 2
I have reviewed this short commentary and think it well-supported, and it addresses a serious issue in the projections of future burned area. This is an important message to be conveyed. My major suggestion would be that, instead of singling out one study, to acknowledge some of the other studies that have used or discussed SPEI in correlative models to project burned area under climate change (albeit without it being labeled ‘nonstationary’, e.g., Krawchuk et al. 2014, Mann et al. 2016)- and to acknowledge other work that makes similar arguments as here (e.g., Syphard et al. 2018, Bowman et al. 2009). This widens the commentary into a broader statement, and it underlines the importance of it. References at end.
Response: We have now incorporated some of these references. However, we want to point ourt that Mann et al, for instance, actually explicitly introduce some of our arguments (i.e. human related variables) in their models and actually conduct a temporal validation in their study area thus achieving more robustness in their conclusions. In any case, these authors also explicitly point in the direction of extremely high uncertainty in the role of dynamic vegetation-climate factors in determining future fire regimes. Finally, thanks for the suggestion of Syphard et al (2018) as the inclusion of this relevant reference now allows for enhanced generality of the main point introduced in the comment.
Overall, there are some awkward word choices and redundancy, so another edit for language may be merited. It could help tighten up the wording too.
Minor comments:
L15 – and again in introduction, certainly there are most likely to be negative feedbacks, but there could be other effects or feedbacks, depending on direct and indirect climatically or fire-driven changes due to CO2, direct species distribution changes, species response to fire, etc. Maybe just say “changes in vegetation” here, then explain as you already do in introduction.
Also, just a note, but the abstract and beginning of introduction are almost verbatim the same. Maybe flesh out introduction a bit more, depending on word limits?
Response: we have now stressed both positive and negative feedbacks. The abstract has now been rewritten.
L14, It took me a while to stop having to look up what SPEI meant when I read it. Might be more intuitive to just call it “fire-drought” or “fire-evapotranspiration”?
Response: We used now fire-drought for clarity and except in the case in which we refer to the burnt area – evapotranspiration coefficients estimated by Turco et al. to account for the fire-drought relationships.
L31, Perhaps start with a background statement (or s) like in the abstract.
Response: Done.
L62, What is PRE-PET?
Response: These are water related variables. We have used now this more general term in order to enhance readability
L86, In this paragraph, which I totally agree with, perhaps mention that human influence is more than just management? fire ignition patterns may also come into play and are not accounted for in the correlative models.
Response: Reference to ignition patterns added.
Mann M, Batllori E, Moritz M, Waller E, Berck P, Flint A, et al. Incorporating anthropogenic influences into fire probability models: effects of human activity and climate change on fire activity in California. PLoS One. 2016;11(4):e0153589.
Krawchuk MA, Moritz MA. Burning issues: statistical analyses of global fire data to inform assessments of environmental change. Environmetrics [Internet]. 2014;25(6):472–81.
Bowman DMJS, Balch JK, Artaxo P, Bond WJ, Carlson JM, Cochrane MA, et al. Fire in the Earth System. Science (80-) [Internet]. 2009;324(5926):481–4.
Syphard, Alexandra D., et al. "Mapping future fire probability under climate change: Does vegetation matter?." PloS one13.8 (2018): e0201680.